# The Perception and Impact of Relative Value Units (RVUs) and Quality-of-Care Compensation in Neurosurgery: A Literature Review

**DOI:** 10.3390/healthcare8040526

**Published:** 2020-12-01

**Authors:** Praveen Satarasinghe, Darsh Shah, Michael T. Koltz

**Affiliations:** 1Department of Neurosurgery, Dell Medical School, 1501 Red River Street, Austin, TX 78723, USA; praveensatarasinghe@utexas.edu (P.S.); darsh.shah@utexas.edu (D.S.); 2Department of Neurosurgery, Seton Brain and Spine Institute, 301 Seton Parkway, Round Rock, TX 78665, USA

**Keywords:** quality of care, relative value units, value-based care, fee-for-service

## Abstract

The debate surrounding the integration of value in healthcare delivery and reimbursement reform has centered around integrating quality metrics into the current fee-for-service relative value units (RVU) payment model. Although a great amount of literature has been published on the creation and utilization of the RVU, there remains a dearth of information on how clinicians from various specialties view RVU and the quality-of-care metric in the compensation formula. The aim of this review is to analyze and consolidate existing theories on the RVU payment model in neurosurgery. Google and PubMed were searched for English-language literature describing opinions on the RVU in neurosurgery. Commentary was noted to be primary opinions if it was mentioned at least twice in the eight articles included in this review. Overall, seven primary opinions on the RVU were identified across the analyzed articles. Integration of quality into the RVU is viewed favorably by neurosurgeons with a few caveats and opportunities for further improvement.

## 1. Introduction

In 2018, the United States spent USD 3.6 trillion or 17.7% of its gross domestic product (GDP) on healthcare expenditures. It is projected that national healthcare spending will reach USD 6.2 trillion in 2028 or 19.7% of GDP. Projections estimate that healthcare spending will grow 1.1% faster than GDP each year from 2019 to 2028 [1]. Many blame the traditional “fee-for-service” payment model for the excessive and rapidly escalating healthcare costs in the United States. Thus, to attempt to control healthcare spending and improve the quality of patient care, payment models are transitioning from traditional volume-driven, fee-for-service reimbursement to value-based payment systems.

Relative value units (RVUs) is a payment model utilized by both the Center for Medicaid and Medicare Services (CMMS) and private insurers to determine how physician services translate into value and ultimately compensation. The RVU is used to determine monetary value for services using a formula accounting for (1) work (wRVU), (2) practice expenses (peRVU), and (3) malpractice RVUs (mRVU) [2]. However, compensation under this current model rewards work volume and does not take into account patient outcomes, delivery of low-cost services, and provision of a better quality of care. Moreover, critics of the current RVU model argue that it punishes coordinated and empathetic care, devalues the patient/caregiver experience, and ultimately leads to the mistaken conclusion that quantity is synonymous with efficiency [3,4,5].

Quality integration into the RVU compensation model has been heavily discussed in the literature. Giacoma et al. designed and quantified a compensation model for transplant surgery that incorporated non-billable, value-generating work—dubbed customized RVU (cRVU)—into the existing RVU compensation model. The authors calculated cRVU for value-generating activities—such setting up satellite and virtual clinics for patients in need, improving quality and safety processes in their practice, coordinating patient care with other providers, and enrolling patients in research studies and clinical trials—into the existing RVU compensation model [6]. Similarly, authors representing a variety of medical specialties including radiology, vascular surgery, and gastroenterology have advocated for integration of desired outcomes such as quality of care, patient outcomes, patient satisfaction scores, teaching, and research into the existing RVU model [7,8,9].

Even in neurosurgery, where RVUs are performed and compensation for RVU is high compared to other medical and surgical specialties, there has been a push towards integration of quality-of-care measurements [10]. In this article, the authors systematically analyzed the current literature for opinions on the RVU payment systems in neurosurgery. Although there is a great amount of literature published on the creation and utilization of the RVU, there is a dearth of information on RVU perception and the quality-of-care metric in the compensation formula by specialty—one of them being neurosurgery. Through a review of the literature, the authors sought to analyze and consolidate existing theories on the RVU payment model within neurosurgery.

## 2. Materials and Methods

### 2.1. Overview

The authors reviewed all of the current literature regarding the RVU and neurosurgery. Two methods of literature collection were utilized in this study: (1) PubMed and (2) Google. Both sources were utilized to ensure comprehensiveness of the literature review.

### 2.2. Search

First, the PubMed database was used to find articles relevant to the study using the search terminology “(neurosurgery) AND relative value unit” or “(neurosurgery) AND RVU”. Second, the same phrases of “neurosurgery AND relative value unit” or “neurosurgery AND RVU” were entered in the Google search engine to find articles that may have been missed via the PubMed search. The Google search was completed on an incognito window after clearing previous search history.

### 2.3. Inclusion Criteria

All articles, irrespective of publication date, were considered. Studies were excluded if (1) full text was unavailable, (2) text was not written in English, (3) text did not mention both neurosurgery and RVUs somewhere, or (4) text was deemed irrelevant by authors.

### 2.4. Full-Length Article Criteria

After applying the search strategy and filtering the considered 22 articles with the inclusion criteria, 8 full-length articles (Table 1) that included discussion of RVUs in neurosurgery were identified and read by both the primary and senior authors. Findings were extracted from the articles and summarized as primary opinions on the RVU in neurosurgery. Commentary was noted as a primary opinion if it was mentioned in at least 2 of the 8 articles included in this review.

## 3. Results

After thorough literature review, seven primary opinions on the RVU in neurosurgery were identified across the analyzed articles. These opinions include the belief among neurosurgeons that RVU compensation should incorporate factors such as quality of care, safety, productivity, and performance; RVU payment schematics should be practice dependent; RVU reimbursement rates are among the highest in neurosurgery; continual participation in Medicare without RVU adjustment will be harmful to their practice in the long run; RVU payment models differ geographically; an emphasis on generating a greater quantity of RVUs has forced them to relay many routine responsibilities to other medical staff; and operative time should be factored into RVU compensation calculations. The results of this literature review are summarized below (Table 2).

## 4. Discussion

The transition to value-based care and the integration of quality-of-care metrics into an existing fee-for-service model is a challenging and longitudinal task that requires further investigation. As evidenced in the results from our literature review, there is still a lot to be discovered on neurosurgery opinions regarding RVU payment models. Furthermore, neurosurgeons do not completely agree on a complete switch to a healthcare payment system focusing on quality over quantity. However, there are seven primary opinions or common themes voiced in the literature with regards to the state of RVU in neurosurgery.

First, there is indication in the literature that neurosurgeons want some form of quality-of-care integration into compensation models. With a large proportion of neurosurgeons making their salaries from Medicare patients, employing hospital systems are starting to offer incentives based on quality, patient satisfaction, and administrative responsibilities [11]. The favoring of some form of quality metric also stems from a desire to combat a “bait and switch” employment philosophy [10]—when an established neurosurgeon is replaced with a less compensated neurosurgeon (who can complete a similar quantity of tasks) without regards to important factors such as stability, quality, and employee loyalty. More recently, neurosurgeons have also been fond of bundled payments that offer compensation based on an entire episode of care rather than a set number of RVUs collected, allowing for quality improvement, reduction in costs, and reduction in adverse events [12]. In the bundle payment system, neurosurgeons are still reimbursed in a similar way for work performed, though financial incentives exist for patient outcomes.

Second, a complete switch to quality-based payment models in neurosurgery is not realistic given the existence of different care models. Rather, neurosurgeons believe that quality integration should be practice-dependent. For example, call schedules for neurosurgeons can be drastically variable. Some hospitals may have heavy calls while others have call shortages. As a result, certain neurosurgeons are unfairly impacted from a payment system based on total RVUs, which is dependent on patients seen during call. In neurosurgical practices with low call schedules, greater integration of quality-of-care metrics may be beneficial and provide a method of standardized valuation [13].

Third, neurosurgeons are aware that their field has one of the highest RVU reimbursement rates. Median RVU compensation for neurosurgeons, regardless of procedure, is USD 71.81 per RVU, the highest among surgical specialties and the second highest behind hematology/oncology specialists [11,14]. Even with concern among neurosurgeons that a quality-based payment system will fail to provide suitable financial or practical incentives to alter practices, surveys indicate that beyond RVU production, neurosurgeons most value quality measures and patient satisfaction [10,15].

Fourth, and closely related to the above third point, is the belief among neurosurgeons that continued reliance on Medicare without payment model adjustment will be harmful to their practice in the long run. Because more than a third of neurosurgeons’ patient population participates in Medicare, 96.8% of neurosurgeons participate in Medicare reimbursement [12]. However, there has been a recent push to avoid Medicare patients due to fixed RVU payment reimbursement models and Medicare payments restricted by factors such as diagnoses and appointment slots. To avoid this decline, the integration of a quality metric into RVU models may offer a solution. Quality adjustment can compensate for the restriction on quantity set by Medicare, and the impetus already exists: since the Medicare Access and CHIP Reauthorization Act (MACRA) of 2015, Medicare has been pushing for integration of quality and value into payment models [16].

Fifth, RVU payments for neurosurgeons differs by geography. Geographic Pricing Cost Index (GPCIs) are meant to adjust for geographic variation in inputs and outputs (e.g., equipment, supplies, costs, population) [17]. Among physicians in private practice, there are mixed reviews regarding GPCIs because of the belief that income may not be accurately described by the total composite of services provided. A quality metric may offer another missing piece of the payment puzzle. Though quality of care is dependent on materials such as supplies and equipment, quality of care is also dependent on physician characteristics and behaviors. This feature, regardless of geography, has the opportunity to be fairly consistent among neurosurgeons.

Sixth, quantity-over-quality pressures from the current RVU system have burdened other medical staff. With an inability to escape RVU-producing tasks, neurosurgeons rely on medical students, residents, and nursing staff to absorb routine responsibilities [18]. Inter-professional collaboration in healthcare is of upmost importance and has been shown to improve patient outcomes [19]. Moreover, specifically in neurosurgery, one retrospective cohort study exploring outcomes of task sharing between general surgery residents and neurosurgeons found that when it came to managing basic emergency neurosurgical care, the resident cohort achieved similar outcomes to their neurosurgery mentors [20]. Through reduction in the quantity burden with incorporation of quality, neurosurgeons and the remainder of the healthcare team will be benefited.

Seventh, and lastly, neurosurgeons may benefit from an RVU payment model based on complexity of procedure. When assessing quality-of-care in surgery, it is sometimes difficult to find an appropriate measurement scale [14,21]. Quality metrics should be based on patient outcomes and effort required to provide a procedure or service. In such a light, more complex and technically taxing procedures should be compensated more favorably. This opinion among neurosurgeons represents the belief in the literature that quality can be represented in many ways.

In summary, there is a still a lot to be uncovered regarding the RVU payment model in neurosurgery. The quality integration reform takes patience and time, with many factors that need to be considered to change the mindset of healthcare providers. However, the seven primary opinions on the RVU in neurosurgery identified in this review of the literature do indicate that a quality metric is viewed favorably by neurosurgeons with a few caveats and opportunities for further improvement. Regardless of the specialty, the entire process of healthcare should be rewarded, not only the results. Testing of different RVU payment models in neurosurgery and further investigation of opinions among neurosurgeons will shed light on the larger goal of integrating quality into the payment model. The end result, as shown by the opinions in the literature, has potential to benefit neurosurgeons, patients, the medical team, and the overall healthcare ecosystem.

## 5. Conclusions

Value-based healthcare involves delivering higher quality care at a lower cost. With rising healthcare costs in the setting of the traditional fee-for-service RVU payment model, value-based payment reform has been gaining traction within many medical specialties. In neurosurgery, viewpoints in the literature parallel the shift towards a quality-centric payment system, as evidenced by the seven primary opinions identified in this review. Although there is not universal agreement on implementing a value-based RVU payment system in neurosurgery, current opinions on factors such as RVU reimbursement rates, RVU payment applications, and financial burdens of quantity-centric payments indicate that a value-based RVU system may be in the best interests of both neurosurgeons and patients.

## Figures and Tables

**Table 1 healthcare-08-00526-t001:** Full-length Articles Considered for Analysis.

Lead Author	Article Title	Journal of Publication	Year of Publication
Benzil DL	Defining the Value of Neurosurgery in the New Healthcare Era	Neurosurgery	2017
Rosenow JM	Neurosurgeons’ Responses to Changing Medicare Reimbursement	Neurosurg Focus	2014
Tringale KR	Types and Distribution of Payments From Industry to Physicians in 2015	JAMA	2017
Langdorf MI	Financial Implications for Physicians Accepting Higher Level of Care Transfers	West J Emerg Med	2013
Benzil DL	The Employed Neurosurgeon: Essential Lessons	Neurosurgery	2017
Shenai MB	Assessing the Economic Efficiency of Physician On-call Payments	Cureus	2018
Rapport R	1 and 1 is 11	Neurohospitalist	2012
Orr RD	What provides a better value for your time?	The Spine Journal	2018

**Table 2 healthcare-08-00526-t002:** Primary Opinions on the RVU in Neurosurgery.

Primary Opinions
(1) RVU payment models in neurosurgery should include compensation percentages for factors such as quality-of-care, safety, productivity, and performance.
(2) Different RVU payment schematics can be applied to neurosurgical practices with different care models.
(3) RVU payments by specialty are greatly divergent, with neurosurgery having one of the higher RVU reimbursement rates.
(4) Continued participation in a public hospital payment model driven by Medicare compensation without adjustment of RVU scaling may be threatening to neurosurgeons.
(5) RVU payment models in neurosurgery differ by institution and geographic region.
(6) The burden of producing a higher quantity of RVUs has led to neurosurgeons relaying many of their routine responsibilities to other medical staff.
(7) For neurosurgeons, and other surgeons, RVU calculations by complexity of procedure or average operative time may better serve as a measure of effort compared to number of procedures/services performed.

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
