# Peer review of "The Perception and Impact of Relative Value Units (RVUs) and Quality-of-Care Compensation in Neurosurgery: A Literature Review"

_healthcare, 2020, doi:10.3390/healthcare8040526_

Round 1

Reviewer 1 Report

Interesting article in the context of the US health system, which addresses the lack of research on the perception of UVR and the quality of specialty compensation metrics. I offer some suggestions to improve the article:

1) In lines 66-67 the authors refer to "two methods of literature collection" with rigor, I think they refer to surveys in two databases;

2) In point 2 "Materials and Methods" the number of articles they have achieved with the research should be referred to and only then the number of articles to be analyzed by applying the inclusion criteria;

3) In the discussion they should mention the importance of further research on this subject to increase the evidence and changes may occur in the context of the application of quality parameters in the VUR.

Author Response

Reviewer 1 – Comments Addressed

  • Thank you for pointing this clarification out. The two methods of literature collection referred to in the methods are collection of information for the review of literature from the PubMed database and google.
  • We appreciate this point. The number of articles before filtering for inclusion criteria has been added to the manuscript in line 85.
  • Thank you for bringing this to our attention. We have further emphasized this point in lines 182-184 towards the end of the manuscript.

Reviewer 2 Report

This article summarizes opinions on the RVU payment system in neurosurgery based on a literature review. 

Overall, it is well-written and elaborates on the rationale for transitioning to a value-based RVU compensation plan. 

Comments/questions:

1) Are there any specific examples where a value-based RVU compensation has already been established? Theoretically practical but how is value measured?

2) The N of 7 articles is very limited, are there other specialties that are advocating for this compensation model, if so, is their rationale for transition the same

3) Specific to point #6, the trend of off loading patient care to other medical staff such as hospitalist is universal across surgical specialties. Is there any data or evidence to suggest that they receive higher/lower quality care than if managed by the primary surgical specialty?

4) point #7, I disagree that longer operative times= more complex. There is the factor of surgeon experience and efficiency that plays a role. Longer operative times could potentially translate into increase morbidity

Author Response

Reviewer 2 – Comments Addressed

  • Thank you for this point and we agree with adding information on this subject. The specific examples have been added to the introduction, with discussion of how the value is measured as requested.
  • Thank you for clarifying this point. The number of articles is limited because there is not much information on the compensation model in neurosurgery, hence our emphasis at the end of the manuscript for further studies of investigation.
  • For point 6, we have added some points to justify this, as evidenced in lines 165-168. Thank you for pointing out this important addition.
  • We appreciate this insight. We have clarified this point in the manuscript by highlighting that complexity of the procedure is the main point we are trying to address, with average operative time potentially being a measure of the complexity, not a definite correlation.

Reviewer 3 Report

This article is important in two major respects. First, it addresses the nexus of traditional payment mechanisms and the concepts of quality and value, which is perhaps the most salient issue underlying payment reform efforts. Second, it targets a particular subspecialty – neurosurgery. To be successful, broad-based movements need to understand and somehow “reach” all of the target audiences or intended participants. Sometimes, the lines of communication follow existing patterns, such as by specialty. Applied work such as this literature review might be very helpful to educate the community of neurosurgeons in this important topics of concern.

The methods and evidence base for the article are modest, bordering on informal. The title refers to a literature review and the narrative reads like a large body of literature or even a representative survey of neurosurgeons. In reality, the authors read a handful of articles and noted several “opinions” that were mentioned once or more. Those results constitute the central content of the article. Again, such a small but concentrated roundup of the issues found in a few articles aimed at the target population might serve to inform and possibly activate members of that target population, i.e., neurosurgeons and their clinical teammates. Its interest and importance for broader audiences are questionable.

The article provides extensive descriptive information about the so-called RVU approach to billing and compensation. I would have assumed that most members of the target population would be familiar with these details and the prominence of RVUs in paying for professional services. If the presentation of material about RVUs is informative to the audience, then it could be an opportunity to either defend or critique that approach in the case of neurosurgery, in order to contribute to the overall literature on that subject.

If my assumption is correct that readers will be familiar with RVUs, then it’s an opportunity to reduce that existing content and allocate more to the subject at hand, namely adding a quality dimension to compensation. For example: In what ways do neurosurgery correspond to other subspecialties in key respects? In what ways is it different or even unique? Are the existing or proposed quality measures clinically and scientifically acceptable? How much should quality affect ultimate compensation, in order to drive behavior change or to reward excellence?

Author Response

Reviewer 3 – Comments Addressed

Thank you for all of your thoughtful feedback. The presence of only a few articles is one of the problems we would like to call to attention with this manuscript, emphasized in the discussion that further investigation of payment models in neurosurgery is necessary.

For the extensive billing information, we cut out all of that information and made sure to bring examples from other fields that support and lay a ground for neurosurgery payment models, as evidence in the new introduction. We have also simplified the definition of the RVU and incorporated more case studies in other field to provide relevance to our discussion of value-based compensation in neurosurgery. Your feedback was extremely valuable for making out point better addressed in our revised manuscript.